# Brillouin Scattering Study of Electro-Optic KTa_1−*x*_Nb*_x_*O_3_ Crystals

**DOI:** 10.3390/ma16020652

**Published:** 2023-01-09

**Authors:** Md. Mijanur Rahaman, Seiji Kojima

**Affiliations:** 1Department of Materials Science and Engineering, University of Rajshahi, Rajshahi 6205, Bangladesh; 2Graduate School of Pure and Applied Sciences, University of Tsukuba, Tsukuba 305-8573, Japan

**Keywords:** perovskite, ferroelectric, polar nanoregions, random fields, macro- and nanodomains, Brillouin scattering

## Abstract

The functionality enhancement of ferroelectrics by local polar clusters called polar nanoregions (PNRs) is one of the current interests in materials science. KTa_1−*x*_Nb*_x_*O_3_ (KTN) with perovskite structure is a well-known electro-optic crystal with a large Kerr effect. The existence of PNRs in relaxor-like ferroelectric Nb-rich KTN with homovalent B-site cations is controversial. This paper reviews recent progress in understanding precursor dynamics in Nb-rich KTN crystals studied using Brillouin scattering. The intense central peak (CP) and significant softening of sound velocity are observed above the Curie temperature (*T*_C_) due to the polarization fluctuations in PNRs. The effects of Li-doping, defects, and electric fields on the growth and/or creation of PNRs are found using changes in acoustic properties. The electric-field-induced *T*_C_, which is shifted to higher values with increases in applied voltage, including critical endpoint (CEP) and field gradient by trapped electrons, are discussed as well. This new knowledge may give new insight into advanced functionality in perovskite ferroelectrics.

## 1. Introduction

Lead-based ferroelectrics have been widely utilized in our daily lives owing to their immense piezoelectricity and giant dielectric response; however, lead has adverse environmental effects which limit the future applicability of these materials. Therefore, it is urgent to develop lead-free ferroelectric materials with excellent functionality for materials science and engineering. The role of local polar clusters called polar nanoregions (PNRs) is crucial to enhance the functionality of lead-free ferroelectric materials [1]. The oxide ferroelectrics with ABO_3_-type perovskite structure are one of the most interesting research fields to explore the role of PNRs in relation to relaxor ferroelectrics (RFEs) and their phase transitions. Amid RFEs, the lead-free relaxor-like ferroelectric potassium tantalate niobate (KTa_1−x_Nb*_x_*O_3_, KTN) with no charge disorder by homovalent occupancy of B-site cations can be a suitable candidate for investigating dynamic anomalies related to PNRs, because of its structural simplicity in comparison with lead-based RFEs such as Pb(Mg_1/3_Nb_2/3_)O_3_ (PMN) and Pb(Zn_1/3_Nb_2/3_)O_3_ (PZN) with charge disorder by heterovalent occupancy of B-site cations. KTN is the solid solution of quantum paraelectric potassium tantalate (KTaO_3_, KTO) and classical ferroelectric potassium niobate (KNbO_3_, KNO). The ferroelectric phase transition of one of the end member KTO at a very low temperature is suppressed by quantum fluctuations [2,3], whereas another end member KNO displays normal ferroelectric nature and undergoes successive phase transitions from the rhombohedral (R) to orthorhombic (O), tetragonal (T), and cubic (C) phases on heating from a low temperature [4,5]. Moreover, the ferroelectric phase transitions and the associated optical properties of KTN can be customized by changing the Nb content [6,7]. In addition, KTN has drawn much scientific attention because of its excellent quadratic electro-optic (EO) coefficient and large photorefractive effect in the vicinity of the Curie temperature (*T*_C_) [8,9,10,11]. Different types of optical devices such as varifocal lenses, modulators, deflectors, and EO switches based on KTN crystals have been developed because of their giant EO effect [12,13,14,15].

RFEs are delineated by the random fields (RFs) that induce the remarkable frequency dispersion and the broad peak in the temperature dependence of the dielectric permittivity. It is well established that such peculiar properties are attributed to RFs related to PNRs [16]. The relaxor nature of ferroelectrics stems from the structural disorder [17,18], which attributed to the random occupation of various types of ions (e.g., Mg^2+^ and Nb^5+^ in PMN; Zn^2+^ and Nb^5+^ in PZN) on a crystallographically equivalent site. Their heterovalent random occupancy is a common source of structural disorder of RFEs [17,18], which can lead to the creation of chemical order regions (CORs) and RFs associated with PNRs. The most important feature of KTN is the off-center displacement of Nb ions that occurs at the B-site and its microscopic origin was rationalized by the pseudo-Jahn–Teller effect (PJTE) [19]. Therefore, the formation of PNRs in KTN stems from the off-center displacements of Nb ions at the B-site, which lead to the relaxor-like nature. On the other hand, the ordered regions of KTN are distinct from those of PMN and PZN, because there is no COR in KTN that induces the diffusive phase transition of a Pb-based RFE with the heterovalent cations at the B-site [20]. Using X-ray absorption fine-structure (XAFS) measurements, the off-center displacements of Nb ions at the B-site of KTN were precisely determined by Hanske-Petitpierre et al. [21]. The off-centering of Nb ions in KTN was also discussed with the eight-site model where Nb ions displace along the eight equivalent [111] directions [22]. The study of acoustic emission of a high-quality KTN (*x* = 0.32, KTN/0.32) single crystal revealed its relaxor-like nature by observing two characteristic temperatures, namely the Burns temperature, *T*_B_~620 K, and the intermediate temperature, *T**~310 K [23]. Such a relaxor-like nature was also found in normal ferroelectric BaTiO_3_ [24]. The distinct types of experimental studies such as Brillouin scattering [25,26,27], Raman scattering [28,29,30,31,32], dielectric spectroscopy [33], refractive index, linear birefringence [34], an ultrasonic pulse-echo technique [35], and infrared absorption [36] revealed the presence of PNRs in a paraelectric cubic phase of non-doped KTN and Li-doped KTN crystals. However, the dynamic behavior and microscopic origin of PNRs still have a long way to apprehend.

Nowadays, the study of Li-doping to KTO and KNO is one of the active research areas in materials science and condensed matter physics. The ferroelectricity appeared and quantum paraelectricity was suppressed with Li-doping to KTO. The substitutional Li ions in K_1-*y*_Li*_y_*NbO_3_ (KLN) and K_1-*z*_Li*_z_*TiO_3_ (KLT) take up one of six off-center sites along the [100] directions at the A-site [37,38], which can lead to the creation of RFs that enhance the appearance of PNRs. However, there is a phase boundary of KLT at about *z* = 0.022, below which KLT freezes into a dipole glass state at a glass transition temperature, *T*_g_, and above which it undergoes a ferroelectric phase transition at the *T*_C_ [37]. Moreover, the KLT/0.026 (*z* = 0.026) showed a relatively broad phase transition, while KLT/0.063 (*z* = 0.063) exhibited a sharp phase transition under a moderate electric field along the [100] direction [39]. Zhou et al. [40] reported that the electromechanical coupling constant, *k*_31_, and the piezoelectric coefficient, *d*_33_, were enhanced up to 40.4% and 431 pC/N, respectively, by Li-doping to KTN. Li et al. [41] also studied the shear components of the piezoelectric response, *d*_31_, and the pyroelectric coefficient of high-quality Li-doped KTN single crystals to apply these materials in pyroelectric and piezoelectric applications such as actuators, sensors, and transducers. Therefore, the studies of electric field and Li-doping effects on the precursor dynamics in the vicinity of the ferroelectric phase transition temperature are crucial for accelerating the understanding and application of these materials.

It is also significant that the structure and physical properties of non-doped KTN and Li-doped KTN are strongly dependent on their chemical compositions [6,7]. Among the various physical properties, the elastic stiffness constants, *C*_ij_ (i,j = 1~6), which are closely related to the phase velocities of acoustic phonons at the Brillouin zone center and inter-atomic potential, play a vital role in materials science. The ultrasonic pulse-echo technique revealed that the elastic constants gradually decrease with an increase in Nb content [35]. Similar results were also found in KTN (0.003 ≤ *x* ≤ 0.057) studied by Rytz et al. [42]. Using Brillouin scattering, the change of elastic constants was found in the case of Pb[(Zn_1/3_Nb_2/3_)*_x_*Ti_1−x_]O_3_ (*x* = 0.09, 0.045) [43]. In the cubic phase of KTN, the investigation of temperature-dependent elastic constants denoted a strong coupling between PNRs and acoustic phonons [27,35,44], but the composition gradient of these physical constants is rather scarce. The value of *T*_C_ in the vicinity of room temperature (RT) is desirable for integrated optics; thus, it is important to investigate the composition gradient of KTN in the range where *T*_C_ is about RT. Additionally, defects significantly affect the elastic properties of materials. A theoretical study of the defect-dependent elastic properties of silicon revealed that both vacancy and interstitial defects, random and singly in combination, decrease the elastic constants *C*_44_ and *C*_11_ [45]. Burnett and Briggs also observed the decrease in elastic constants *C*_11_ and *C*_44_ of silicon with the increase in defects [46]. To the best of our knowledge, the effects of defects on the elastic properties of high-quality non-doped KTN and Li-doped KTN are rare.

Recently, space-charged controlled electro-optic (EO) devices such as optical beam deflectors, which are applied in light detection and ranging (LiDAR) systems and laser displays, are interesting topics in applied physics. The space charge in KTN crystals is formed due to the injection of electrons from the cathode with the application of a voltage to the electrode pairs [15,47]. The phenomenon of optical beam deflection by space charge was found in KTN crystals [15,47]. Thus, the space charge creates the distribution of electric fields at first; after that, the refractive index distribution occurs because of the EO effect. This refractive index distribution bends a light beam, and the deflector may achieve deflection angles exceeding several degrees, which is much higher than those observed from the conventional EO effect [48]. Interestingly, the injected electrons are stabilized in KTN crystals trapped by the localized states. Nakamura et al. [15] found the effect of the space charge on the optical beam deflection and observed that the electric fields induced by space charges are not uniform inside a KTN crystal. On the contrary, the injected electrons remarkably increase the capacitance and permittivity of a Li-doped KTN crystal [49]. It was reported that the critical endpoint (CEP) is the driving mechanism of polarization rotation and is associated with colossal electromechanical response in RFEs. The electric-field–temperature phase diagram including the CEP was reported for Li-doped KTN crystals without electron injection [49]. However, the investigation of the effects of injected and/or trapped electrons on the PNRs and CEP of non-doped KTN and Li-doped KTN crystals is deficient, which is crucial for application to electromechanical devices.

In the present study, we review the recent extensive studies on precursor dynamics in KTN crystals using inelastic light scattering spectroscopy. The effects of Li-doping, defects, composition gradient, and injected electrons on PNRs of KTN crystals are discussed as well. The acquired knowledge, which provides new insights into the various dynamics of PNRs, may help to enhance the functionality of ferroelectric materials.

## 2. Methodology: Micro-Brillouin Scattering

The elastic properties, including different types of elastic moduli and acoustic absorption coefficients, are one of the fundamental properties of a material. The elastic constants (*C*_ij_) of a material, which are directly related to the interatomic forces in the crystal, can be determined using the velocity of the sound wave related to the acoustic phonons and density of the material. There are different techniques to estimate the value of elastic constants. Among them, the mechanical resonance and the ultrasonic pulse-echo techniques have been well established [35,50,51]. However, Brillouin spectroscopy, which covers the range of 1~1000 GHz frequency, is a powerful, non-destructive, non-contact tool to measure elastic constants under various external conditions such as temperature, pressure, and electric fields. Brillouin scattering has been utilized to study not only acoustic phonons but also the low-frequency optical soft mode [52], magnon [53], fracton [54], and soliton [55].

The optical arrangement of a typical Brillouin scattering experiment, which was developed by Sandercock [56], uses a combination of two Fabray–Pérot interferometers (FP1 and FP2) (JRS TEP-1, JRS, Zwillikon, Switzerland) with slightly different free spectral ranges as shown in Figure 1. Laser light is applied to the optical microscope with the help of a small mirror to focus the samples. The scattered light is gathered using the objective lens and entered into the FP1 through the aperture A1 and adjustable pinhole P1. The mirror M1 reflects the light toward the lens L1, which redirects it via mirror M2 to FP1 after the collimation. The deflected light from M2 is then passed through puncture 1 of the mask A2 and is directed to FP2 via mirror M3. The light hits the 90° prism PR1 after transmission through FP2, is reflected downwards, goes back parallel to itself towards FP2, and continues to FP1 through aperture 2 of A2. The light is passed through the lens L1 and underneath the mirror M1 after transmission through FP1 and is focused onto the mirror M4. Mirror M4 redirects the light to FP1 through lens L1 and mirror M2. The combination of mirror M4 and lens L1 lying at its focus is known as catseye and acts as a spatial filter that removes undesirable beams such as the beams reflected from the rear surfaces of the interferometer mirrors. The light is directed to the prism PR2 via the mirror M5 after the final pass through the interferometers. The combination of lens L2, prism PR2, and the output pinhole P2 forms a bandpass filter with a width estimated using the size of the pinhole. The mirror M6 sends the light onto the photomultiplier through the lens L2, aperture A3, and output pinhole P2. Finally, the scattered signal is collected using a conventional photon counting system.

The micro-Brillouin scattering spectra of KTN and Li-doped KTN single crystals were measured using a 3 + 3 pass Sandercock-type tandem Fabray–Pérot interferometer (JRS, TEP-1) equipped with a photon counting system. The KTN and Li-doped KTN single crystals grown using the top-seeded solution growth (TSSG) method at NTT Corporation (Tokyo, Japan) were cut into different dimensions. The largest (100) faces were polished to optical grade to obtain inelastic light scattering spectra. All the Brillouin scattering spectra were recorded at a backward scattering geometry [47]. The Brillouin scattering was excited with a wavelength of 532 nm and a power of about 100 mW from a single-frequency diode pumped solid state (DPSS) laser. The free spectral range (FSR) of the spectrometer was fixed at 300 GHz for the central peak (CP) and 75 GHz for the longitudinal acoustic (LA) and transverse acoustic (TA) mode measurements. The temperature of the sample was controlled with a heating and/or cooling stage (THMS600, Linkam, Salfords, UK,) with the temperature stability of ±0.1 K overall temperature. For applying the electric field, the appropriate surfaces of the sample were coated with silver paste, and gold contact wires were attached to the electrodes.

## 3. Evolution of PNRs: Models

In a paraelectric cubic phase of RFEs, the PNRs are defined by two temperatures, namely the Burns temperature, *T*_B_ [59], below which the dynamic PNRs begin to appear, and the intermediate temperature, *T**, at which the dynamic PNRs start to transform into static and/or rapid growth of these PNRs up to the percolation limit in which the reorientation is completely suppressed [25,26,27,30,60,61]. It is believed that the formation of dynamic PNRs begins at *T*_B_ owing to a short-lived correlation among off-center ions, whereas the dynamic-to-static transition of these PNRs begins at *T** with the sudden growth in size due to the long-lived correlation among the atomic displacements of off-center ions. However, the physical origin and mechanism of the formation PNRs and their dynamic behavior are still a long way from our understanding.

There are various approaches to understanding the formation of PNRs [25,26,27,60,61,62,63,64,65,66,67,68,69]. All of them can be separated into two models suggested by A.A. Bokov and Z.-G. Ye [16]. One model considers that PNRs are a result of local phase fluctuations or local phase transitions, so that the crystal consists of nanosized polar clusters embedded in a nonpolar paraelectric cubic phase in which the symmetry of the lattice remains unchanged [62,63,64,65]. The second model considers the local transition to occur in all the crystal regions containing low-symmetry nanodomains detached not by the regions of cubic symmetry but by the domain walls [66,67]. They also reported that the symmetry of local clusters in the cubic matrix is not expected to be cubic and the domain wall thickness is comparable with the size of nanodomains [16]. However, they did not provide information about the local symmetry of PNRs. Without defining the local symmetry of polar clusters in RFEs, J. Toulouse [68] suggested that the static PNRs’ so-called polar nanodomains (PNDs) appear at *T** upon cooling. He also suggested that PNDs (static PNRs) represent the permanent polarization appearing with local strain fields as PNDs do not mean that they are “frozen” or inert, but their reorientation may contribute to the relaxor nature below *T** in RFEs. The reorientation and vibrational-field-driven dynamics of PNRs in the vicinity of the *T*_C_ of relaxor-like KTN crystal were studied by Tan et al. [70]. The neutron pair distribution function analysis revealed local rhombohedral *R3m* symmetry of the PNRs of a PMN single crystal [71]. Taniguchi et al. [72] reported the local cubic $Fm\bar{3}m$ symmetry for the COR and rhombohedral *R3m* symmetry for the PNRs in the paraelectric cubic phase of PMN single crystal using Raman scattering. They also reported that the local symmetry breaking due to the existence of PNRs in the cubic phase of the PMN with heterovalent B-site cations stems from the *E*(*x,y*) mode of rhombohedral *R3m* symmetry [72]. Recently, Helal et al. [69] reported that the PNRs of PMN-0.56PT (*x* = 0.56) with weak RFs belong to the local tetragonal symmetry with the point group *4mm*, which is distinct from local rhombohedral *R3m* symmetry of PMN with strong RFs.

Recently, Dey et al. [73] visualized randomly oriented nanodomains within macro-domain states in the ferroelectric phase of the (1 − *x*)Ba(Zr_0.2_Ti_0.8_)O_3_ − *x*(Ba_0.7_Ca_0.3_)TiO_3_ (BZT-*x*BCT, *x* = 0.60) ceramics with homovalent cations at both the A- and B-sites. In the ferroelectric phases of the BZT-*x*BCT ceramics, the randomly oriented nanodomains within macro-domain states were also found by observing the prominent CP in Brillouin scattering spectra, which is related to the precursor’s dynamics [74]. Such a type of prominent broad CP, which was attributed to random nanodomain states, was observed in Co-doped BTO ceramics as well [75]. It is well established that the dynamic-to-static transition of PNRs starts at the intermediate temperature, *T**, and most of the PNRs become static in the vicinity of *T*_C_. Thus, it is believed that the static PNRs become randomly oriented nanodomain states due to the freezing out of local polarization in a ferroelectric phase [74,75]. The temperature evolution of PNRs in RFEs with heterovalent cations at the B-site is well explained, but extensive studies on PNRs in relaxor-like ferroelectrics with homovalent cations at B-sites such as KTN are rather scarce. The microscopic scenario of PNRs in KTN may differ from Pb-based RFEs because the PNRs in KTN arise only for the off-centering of Nb ions at the B-site. In the present investigation, the microscopic scenario of PNRs with local structural symmetry including vibrational symmetry in relaxor-like KTN crystals is discussed using a model, as shown in Figure 2, based on the results of inelastic light scattering experiments.

### 3.1. Experimental Evidence of PNRs

The analysis of temperature dependencies of the refractive index, *n*, which is linear at *T* > *T*_B_ but starts to deviate from linearity at *T* < *T*_B_, was the first experimental evidence for PNRs [59]. The deviation of *n* from linearity below *T*_B_ was attributed to the variation in *n* induced by local polarization inside the PNRs via the quadratic EO effect. In RFE PMN, the presence of PNRs was confirmed using synchrotron X-ray and neutron diffuse scattering about the reciprocal lattice points [76,77]. High-resolution neutron elastic diffusion scattering revealed the size of emerging PNRs at about 1.5 nm, which was almost temperature-independent at high temperatures; smaller PNRs reach around 7 nm on cooling [78]. The size of the emergence of PNRs in PMN below *T*_B_ was also observed using transmission electron microscopy (TEM) [79], but their size was larger than that determined from neutron diffuse scattering. The temperature-dependent dielectric properties of PMN were extensively studied to characterize the dielectric relaxation related to PNRs as well [80,81]. However, it is very difficult to record the broadband spectra between 1 to 1000 GHz using conventional dielectric spectroscopic measurements. On the other hand, Brillouin spectroscopy enables the acquisition of such broad spectra with a single scan. Brillouin scattering spectroscopy is a powerful tool to observe the dynamics of local polarization fluctuations in ferroelectrics because it is very sensitive to local heterogeneities associated with compositional and structural disorder among a few techniques that can be used to directly probe the local symmetry and structure. In addition, Brillouin scattering has a spatial resolution comparable to the wavelength of an exciting laser, and it is possible to measure the optic soft mode and acoustic phonons in different types of perovskite ferroelectrics.

### 3.2. PNRs in Relaxor-like Ferroelectric KTN

KTN crystals undergo a successive phase transition of a rhombohedral–orthorhombic–tetragonal–cubic sequence the same as that of KNbO_3_ (*x* = 1.00) for solid solutions of *x* > 0.05 [6,7] on heating, and the phase transition’s temperature can be customized with the Nb content. However, KTN crystals with *x* = 0.008–0.05 undergo only a cubic–rhombohedral phase. Samara [82] reported relaxational glass-like behavior rather than a ferroelectric phase transition for KTN with a small Nb content (*x* < 0.02). Samara also found that, unless subject to high pressure, KTN crystal with *x* = 0.02 did not show any frequency dispersion [82]. On the other hand, refractive index and linear birefringence measurements revealed the diffuse ferroelectric phase transition of KTN/0.02 crystal [34]. The structural phase transition and the existence of PNRs well above *T*_C_ of KTN/0.012 crystal were studied using dielectric, polarization, and Raman measurements [28], which led to the relaxor-like nature of the KTN crystals. The presence of PNRs in KTN/0.09 crystal was also investigated using Raman scattering as well [83]. Using the ultrasonic pulse-echo technique, Knauss et al. [35] found a reduced softening of elastic constants in the vicinity of *T*_C_ in KTN crystals with *x* = 0.012, 0.03, 0.09, and 0.157, which was accompanied by local strain fields related to PNRs that could couple to the sound waves, resulting in a reduced softening of the elastic constants. The relaxor-like nature of a high-quality KTN/0.32 single crystal was seen by observing Burns and intermediate temperatures associated with PNRs at *T*_B_~620 K and *T**~310 K, respectively, using acoustic emission measurement [23]. The relaxor-like dynamics near *T*_C_ of the KTN/0.32 crystal were also investigated using Brillouin scattering [26]. Hence, there is noticeable controversy on the optimum composition at which KTN crystals show a relaxor-to-normal crossover ferroelectric phase transition. In this study, the existence of PNRs and their dynamic nature in Li-doped and non-doped Nb-rich KTN were investigated using Brillouin scattering to clarify the elastic anomaly in the GHz range caused by the coupling of acoustic phonons with the PNRs and their slowing down.

#### 3.2.1. Elastic Anomaly

The measurement of temperature-dependent Brillouin scattering spectra below and above *T*_C_ was carried out to clarify the elastic anomaly in the vicinity of the paraelectric to a ferroelectric phase transition. The Brillouin scattering spectra of high-quality Nb-rich KTN/0.40 single crystal at some selected temperatures measured with FSR = 75 GHz are shown in Figure 3a. As can be seen in Figure 3a, the two doublets and a CP at zero frequency shift appeared in each spectrum in a wide range of temperatures. The low- and high-frequency doublets correspond to the transverse acoustic (TA) and longitudinal acoustic (LA) modes, respectively. It is apparent from Figure 3a that the TA mode persists in the paraelectric cubic phase of the KTN/0.40 crystal, which is forbidden at a backscattering geometry by the Brillouin selection rule [84]. Therefore, the presence of the prominent TA mode in the paraelectric phase may denote the local breaking of cubic symmetry. In the cubic phase of KTN/0.40, the local symmetry breaking owing to the *A*_1_(z) mode of PNRs with rhombohedral *R3m* symmetry was confirmed by the intense first-order Raman mode in the cubic phase that is inactive for the cubic $Pm\bar{3}m$ symmetry by the Raman selection rule [25,85].

The doublets were fitted using the Voigt function to determine the frequency shift and damping of the acoustic phonon modes. The sound velocity (*V*) and sound attenuation coefficient (*α*), which are related to the Brillouin frequency shift ($\upsilon_{B}$) and full width at half maximum (FWHM), (*Γ*_B_), were calculated with the following equations:(1)$$\Delta \upsilon_{B}=\pm \frac{2mV}{\lambda_{0}}sin\left(\frac{\theta }{2}\right)$$
(2)$$\alpha =\frac{\pi \Gamma_{B}}{V}$$
where *m*, *λ*_0_, and *θ* are the refractive index of the observed sample, the wavelength of the incident light, and the scattering angle, respectively. The temperature dependencies of LA and TA velocities of the KTN/0.40 crystal were determined and plotted as shown in Figure 3b,c, respectively. In the cubic phase, the observed LA and TA sound velocities are related to the elastic constants *C*_11_ and *C*_44_, respectively, by the following equations [83]:(3)$$C_{11}=\rho V_{\mathrm{LA}}^{2}$$
(4)$$C_{44}=\rho V_{\mathrm{TA}}^{2}$$

Here, *ρ* is the density of the crystal. The LA velocity exhibits a significant softening near the ferroelectric tetragonal to paraelectric cubic phase transition temperature, *T*_C_ = 308 K, while the TA velocity displays only a modest softening and a small discontinuity at *T*_C_, denoting the behavior of a slightly first-order phase transition [25].

The anomalous changes in the acoustic properties of relaxor-like ferroelectrics can be explained using the Landau theory considering the coupling between strain and order parameters in the free energy expansion. In ferroelectrics, the Landau expression for the free energy, *F,* generally consists of three parts: the first part is related to the order parameter, i.e., polarization, *P,* in the present case; the second part is pertinent to the elastic energy; and the third part is the coupling term between strain and the order parameter. The effective elastic constant *C*_ij_, owing to the variation in polarization, driven by the modification in strain, can be expressed as follows [86]:(5)$$C_{\mathrm{ij}}=C_{\mathrm{ij}}^{\infty }-\underset{k,l}{\sum }\left(\frac{\partial^{2}F}{\partial x_{i}\partial P_{k}}\right){\left(\frac{\partial^{2}F}{\partial P_{k}\partial P_{l}}\right)}^{-1}\left(\frac{\partial^{2}F}{\partial x_{j}\partial P_{l}}\right)$$
where $C_{\mathrm{ij}}^{\infty }$ is the polarization-independent elastic constant, namely, the bare elastic constant in which the temperature dependence is determined by the lattice anharmonicity. There is no macroscopic spontaneous polarization, i.e., <*P*> = 0 in the paraelectric cubic phase of relaxor-like ferroelectric KTN due to the random orientations of PNRs. Hence, we envisage the quadratic piezoelectric coupling between strain and the order parameter would dominate the coupling terms in the free energy below *T*_B_. The variation in elastic constant due to the quadratic piezoelectric coupling can be modelled with the following expression [87]:(6)$$\Delta C=C-C^{\infty }=\gamma \langle P^{2}\rangle \chi$$
where *γ* is the piezoelectric coupling coefficient, $\langle P^{2}\rangle$ is the mean value of the squared local polarization of PNRs, and *χ* is the susceptibility for the order parameter. For the second-order phase transition, the classical Landau theory anticipates only a step-like anomaly for *C*(*T*) at the phase-transition temperature. However, in relaxor-like ferroelectrics the local polarizations emerge at *T*_B_ and they steadily develop with decreasing temperature down to *T*_C_. The appearance of local polarization, although the average polarization is zero, contributes to the decrease in the elastic constants due to the quadratic behavior of the piezoelectric interaction. The decrease in elastic constants evolves steadily without any abrupt change until the local polarizations of PNRs is saturated upon cooling. Hence, the significant softening and the clear anomaly in both the *V*_TA_ (related to *C*_44_) and *V*_LA_ (related to *C*_11_) towards the *T*_C_ is attributed to the coupling of the TA and LA modes with fluctuation of local polarizations in PNRs. Moreover, a clear anomaly and the remarkable increase in the LA mode damping towards *T*_C_ was seen on cooling from high temperature, as shown in the inset of Figure 3d. It is apparent from the inset of Figure 3d that the almost constant temperature-dependent sound attenuation at high temperatures starts to deviate at around 353 K, which was assigned as the intermediate temperature (*T**) at which the rapid growth of PNRs begins, which induces the significant softening of the TA mode as well (Figure 3b). Landau and Khalatnikov [88] performed a mean-field theory of the sound attenuation anomaly and found that this anomaly arises owing to the coupling between the strains and order parameters. Levanyuk et al. [89] observed the temperature-dependent sound attenuation in the following form with the critical exponent of *n* = 1.5 for the order—disorder nature of ferroelectric phase transition based on the mean-field approximation.
(7)$$\alpha =\alpha_{1}{\left(T-T_{0}\right)}^{-n}$$
where $\alpha_{1}$ is the constant. *T*_0_ is the characteristic temperature, in which *T*_0_ = *T*_C_ describes the second-order while *T*_0_ < *T*_C_ explains the first-order nature of the ferroelectric phase transition. The sound attenuation was fitted between *T*_C_ and *T*_C_ + 40 K using Equation (7) (Figure 3d) to explain the anomalous nature and order—disorder behavior of the phase transition. The experimental results were well reproduced by the theory with the critical exponent of 1.5, indicating the order—disorder nature of the ferroelectric phase transition of the high quality KTN/0.40 single crystal. The parameters obtained by fitting the sound attenuation are given in Ref. [23]. Moreover, the good consistency between the observed sound attenuation and the theoretical line indicates that the anomaly arises due to coupling between the LA phonons and polarization fluctuations in the PNRs. These characteristics near *T*_C_ were observed in various lead-based perovskite relaxors such as 0.65Pb(Mg_1/3_Nb_2/3_)O_3_–0.35PbTiO_3_ (PMN–0.35PT), 0.70Pb(Sc_1/2_Nb_1/2_)O_3_–0.30PbTiO_3_ (PSN–0.30PT), 0.85Pb(Zn_1/3_Nb_2/3_)O_3_–0.07PbTiO_3_ (PZN–0.07PT), 0.65Pb(In_1/2_Nb_1/2_)O_3_–0.35PbTiO_3_ (PIN–0.35PT), etc. [84,85,86,87,88,89,90].

#### 3.2.2. Critical Slowing Down

The temperature-dependent broad central peak (CP) measured with FSR = 300 GHz, which is associated with the relaxation process of polarization fluctuations in PNRs, was investigated to clarify the precursor dynamics in KTN crystals. The broad CP of the KTN/0.40 crystal at some selected temperatures is shown in Figure 4a. As can be seen in Figure 4, the CP develops upon cooling and its intensity becomes a maximum at *T*_C_. It is significant that the intensity of the CP drastically increases towards *T*_C_ below *T** as shown in Figure 4b. The increase in the CP intensity upon cooling reflects the increase in the correlation among PNRs. The anomalous changes in the CP intensity and optical Raman modes of PZN crystals revealed that *T** exists between *T*_B_ and *T*_C_ [60]. They also reported that the temperature region between *T** and *T*_B_ is characterized by the formation of short-lived correlations between the off-centered ions resulting in the formation of small dynamic PNRs, while the region between *T*_C_ and *T** is characterized by coupling between adjacent PNRs and their aggregation to form larger PNRs [60]. Such anomalous variation of optical Raman modes and CP intensity was also observed in a KTN/0.40 crystal using Raman scattering [85]. Hence, the significant increase in damping of the LA mode below *T** indicates the rapid growth of PNRs, which also induces a remarkable increase in the CP intensity and a significant softening of *V*_TA_ [25,26,91,92,93,94].

The phenomenon of critical slowing down is known as the order–disorder nature of a structural phase transition. The relaxation time of the polarization fluctuations related to ferroelectric phase transitions was found using broadband Brillouin scattering spectroscopy [92,93]. The relaxation time was determined using the width *Γ*_CP_ of the CP using the following equation [93].
(8)$$\pi \Gamma_{\mathrm{CP}}\tau_{\mathrm{CP}}=1$$
where $\Gamma_{\mathrm{CP}}$ and $\tau_{\mathrm{CP}}$ are the width of the CP and relaxation time, respectively. The inverse relaxation time as a function of temperature (Figure 4c) exhibits a linear dependence between *T*_C_ and *T*_C_ + 49 K, which denotes the fact that a paraelectric cubic to ferroelectric tetragonal phase transition is the order–disorder type. The critical slowing down near *T*_C_ is given by
(9)$$\frac{1}{\tau }=\frac{1}{\tau_{0}}+\frac{1}{\tau_{1}}\left(\frac{T-T_{C}}{T_{C}}\right)$$

$\tau_{0}$ and $\tau_{1}$ are the relaxation time and the characteristic time, respectively [25,26,95]. The observed values of $\tau_{0}$ and $\tau_{1}$ are 1.61 ps 14.8 ps, respectively, in the temperature range between *T*_C_ and *T*_C_ + 49 K [25]. It is significant to compare the value of $\tau_{0}$ with a typical relaxor system that showed the order–disorder nature of the ferroelectric phase transition near *T*_C_. The observed value of $\tau_{0}$ of the KTN/0.40 crystal is comparable to that of the PZN–0.15PT crystal [96]. Hence, one can conclude from these results that an order–disorder mechanism in the KTN/0.40 single crystal contributes to slowing down the dynamic PNRs.

For a better understanding of the temperature evolution of the dynamic PNRs, the average size of the dynamic PNRs, *l*_PNRs_, was determined using the equation $l_{\mathrm{PNR}}=V_{\mathrm{LA}}\times \tau_{\mathrm{CP}}$, because the maximum of the characteristic length of the strain fluctuations is given by the propagation length of strain within the relaxation time [97]. It is apparent that the average size of the dynamic PNRs increases towards *T*_C_ upon cooling, as shown in Figure 4d, indicating the growth of the PNRs up to the percolation limit in which the reorientation is completely suppressed [61]. The average size of PNRs is about 4.5 nm at an intermediate temperature, *T**, which is in agreement with the size of a dynamic polar cluster of 4–6 nm [26].

### 3.3. Li-Doping Effects on PNRs in KTN

The lattice dynamic properties related to a relaxor ferroelectric phase transition of high quality 2.5%Li-doped KTN (KLTN/0.025/0.38) and 5%Li-doped KTN (KLTN/0.05/0.27) single crystals were investigated and compared to the results of the non-doped KTN/0.40 crystals to understand the effects of Li-doping on PNRs. The observed Brillouin scattering spectra of the KLTN/0.05/0.27 single crystal at some selected temperatures are shown in Figure 5a. The Brillouin scattering spectra of the KLTN/0.05/0.27 crystal are qualitatively similar to spectra of a high-quality KTN/0.40 single crystal with no Li content. The elastic constant *C*_11_ was calculated from the sound velocities with Equation (3) using a density of *ρ* = 6630 kg/m^3^ for Li-doped KTN [40] and *ρ* = 6078.65 kg/m^3^ for non-doped KTN/0.40 [98]. The estimated elastic constant *C*_11_ of non-doped and 5%Li-doped KTN crystals as a function of temperature is displayed in Figure 5b. It is seen that the elastic constants of non-doped KTN are smaller than those of the Li-doped KTN crystals in the studied temperature range. However, it is found from Figure 5b that Li-doping broadens the temperature region of the elastic anomaly near the *T*_C_. The broadening of the temperature region may be caused by the growth of PNRs induced by Li ions. Hence, the elastic anomaly broadening near *T*_C_ of Li-doped KTN crystals in comparison to that of the non-doped KTN can be an indication of the expansion of the temperature region of the elastic anomaly [27].

We also studied the frequency and damping of the acoustic phonons (Figure 5c,d to clarify the extension of the temperature region by Li-doping. It was challenging to make decisive data of the LA mode damping on fitting spectra using Lorentzian functions because of its asymmetric nature (Figure 5a). Thus, we concentrated on the frequency and damping of the TA mode. It is apparent from Figure 5c,d that the temperature interval between *T** and *T*_C_ was enhanced by the Li-doping. As can be seen in Figure 5c, the attenuation of the TA mode remarkably increases with Li-doping, which means the TA phonons were markedly scattered by the Li-doping, resulting in an increase in the damping and decrease in the frequency of the TA phonons. These results are a clear indication of enhanced growth of PNRs by Li-doping. Hence, the expansion of the temperature interval between *T*_C_ and *T** can be attributed to Li-ion-induced enhanced stability of PNRs. With Li-doping, the enhanced stability of PNRs persuades the elastic anomaly broadening, which can be an indication of the enhanced relaxor nature of KTN crystals with Li-doping. A comparative study of the dielectric properties among non-doped, 2.5%Li-doped, and 5%Li-doped KTN crystals was performed to confirm the enhanced relaxor nature of KTN with Li-doping. The dielectric constant follows the well-known Curie–Weiss law 1/*ɛ* = (*T* − *T*_C_)/*C* above the *T*_C_ in the case of normal ferroelectrics, where *C* is the Curie constant, whereas this law is obeyed only at temperatures higher than the *T*_B_ in the case of RFEs [79,80]. The dielectric constant below *T*_B_ in the paraelectric phase of the relaxor-like ferroelectrics can be discussed with the following equations:(10)$$\frac{1}{\varepsilon }=\frac{1}{\varepsilon_{m}}\left[1+\frac{{\left(T-T_{m}\right)}^{\gamma }}{2\delta^{2}}\right],\left(T>T_{m}\right);1<\gamma \leq 2$$
where $\varepsilon_{m}$ denotes the magnitude of the dielectric peak at the *T*_m_. *δ* and *γ* are fitting parameters denoting the degree of the diffuseness of the phase transition. When *γ* = 2 it describes a typical relaxor property, while at *γ* = 1, Equation (10) is the same as the Curie–Weiss relation. In RFEs, such a peculiar property is attributed to RFs related to PNRs. It was found that the inverse of the dielectric constant of non-doped, 2.5%Li-doped, and 5%Li-doped KTN crystals is well fit by Equation (10) with the critical exponent γ of 1.12 ± 0.006, 1.19 ± 0.003, and 1.31 ± 0.005, respectively, indicating a relaxor-like nature of the non-doped KTN and the Li-doped KTN crystals [58,99]. It is significant that the values of fitting parameters gradually increase with the Li content, which confirms the Li ions induced enhanced relaxor natures in KTN crystals.

To explain the Li-doping effects on elastic anomaly caused by PNRs, the elastic constant *C*_11_ curves of the KTN/0.40 and KLTN/0.05/0.27 single crystals (Figure 5b) were fit with the following equation [35,100,101]:(11)$$C_{11}=A_{1}+A_{2}T-A_{3}{\left(T-T_{0}\right)}^{-q}$$

The second term describes the anharmonic behavior. The critical softening owing to polarization fluctuations in PNRs is described by the third term, where *A*_3_ denotes the elastic anomaly. The value of the critical exponent *q* was predicted to be 0.5 for three-dimensional fluctuations, 1.0 for two-dimensional fluctuations, and 1.5 for one-dimensional fluctuations [102]. A good consistency between the fitting and experimental results was obtained with an exponent of *q* = 0.5 reflecting the three-dimensional fluctuations of polarization in a cubic phase of non-doped and Li-doped KTN crystals [27]. It is significant to compare the value of the anomalous part, i.e., *A*_3_, of the non-doped KTN (*A*_3_ = 242.55 ± 8.49) and 5%Li-doped KTN (*A*_3_ = 358.92 ± 10.50) crystals [27]. It is clear that the magnitude of the anomalous part was significantly increased by the Li-doping. The remarkable increase in the amplitude of the elastic anomaly in Li-doped KTN is clear evidence of the expansion of the temperature region of the elastic anomaly attributed to PNRs in comparison to that of the non-doped KTN.

In order to have a better understanding Li-doping’s effect on PNRs, the comparative study of the CP behavior as a function of temperature was performed, which is related to the relaxation process in fluctuating PNRs. The CP spectra of the KTN/0.40 and KLTN/0.05/0.27 single crystals measured at 353 K with FSR = 300 GHz are shown in Figure 6a. It was seen that the intensity of the CP significantly increases with Li-doping, as shown in Figure 6a,b. The remarkable increase in intensity of the CP of the Li-doped KTN is owing to the growth of PNRs, which is induced by Li ions. A comparative study of the mean relaxation time, which was estimated by the CP width, between non-doped KTN and Li-doped KTN crystals was also performed to show the Li-doping effects of the precursor dynamics. The temperature-dependent inverse of the relaxation time is displayed in Figure 6c. It was found that the dynamic susceptibility, which was determined from the CP intensity, of the 2.5%Li-doped KTN crystal follows the extended Curie–Weiss law [99]. Since the dynamic susceptibility of the Li-doped KTN crystal follows the extended Curie–Weiss law, the generalized Lyddane–Sachs–Teller (LST) relation anticipates that the inverse of the relaxation time can be explained by the stretched-type slowing near *T*_C_. Under the influence of RFs related to PNRs, such a stretched-type slowing-down behavior of the relaxation time might be described by the following empirical relation [97].
$$\frac{1}{\tau }=\frac{1}{\tau_{0}}+\frac{1}{\tau_{1}}{\left(\frac{T-T_{C}}{T_{C}}\right)}^{\beta },\left(\beta \geq 1\right)\;for\;T>T_{C}$$
where *β* denotes the stretched index associated with the RFs. *β* = 1.0 indicates the normal critical slowing down with almost negligible RFs and it assumes values >1.0 depending on the strength of the RFs. The observed inverse of the relaxation time of 2.5%Li-doped and 5%Li-doped KTN crystals with the critical exponent *β* of 1.10 [99] and 1.18 [58], respectively, indicates the order–disorder behavior of the ferroelectric phase transition with weak RFs.

The average size of the dynamic PNR of the Li-doped KTN crystals was compared with that of the non-doped one to clarify the effects of Li-doping on PNRs. The average size of the dynamic PNR of the non-doped and Li-doped KTN crystals as a function of *T*/*T*_C_ is shown in Figure 6d. It is clear from Figure 6d that Li-doping significantly increases the size of the dynamic PNR. Hence, it is concluded that the intense CP, the broadening of elastic anomaly, and the enhanced relaxor nature of the Li-doped KTN in comparison to that of the non-doped KTN can be attributed to the enhanced growth of PNRs induced by Li ions.

One of the most important effects of Li-doping is the change in the physical origin of the CP of the KTN in the paraelectric cubic phase. It was observed that the CP in the cubic phase of non-doped and 2.5%Li-doped KTN crystals comes from the *A*_1_(z) mode of PNRs [25,85,99], while the CP of the 5%Li-doped KTN stems from the *E*(*x*,*y*) mode of PNRs with a rhombohedral *R3m* symmetry [28] based on angular dependence of Raman scattering spectra with different scattering geometries. Hence, the intense CP of the 5%Li-doped KTN in comparison to the non-doped KTN could be owing to the transformation of the local mode of PNRs in addition to the growth of PNRs. Sokoloff et al. [22] found that the CP intensity caused by *A*-mode fluctuations is less intense than the *E*-mode fluctuations in the tetragonal phase with *P4mm* symmetry of the KTN/0.28 crystal. It is important to explain the microscopic origin of the change in the local mode of PNRs in 5%Li-doped KTN crystals. The change in the local mode of PNRs in 5%Li-doped KTN can be explained by two possibilities:i.In non-doped KTN, the PNRs stem only from the off-center displacements of Nb ions at the B-site along the equivalent [111] directions corresponding to one set of atomic displacements related to 180° fluctuations, i.e., longitudinal polarization fluctuations in PNRs, as shown in Figure 7a [69,103]. Hence, the *A*_1_(*z*) mode of PNRs is reasonable in the case of non-doped KTN crystals. Moreover, it is believed that off-center displacements of Li ions in Li-doped KTN crystals occur at the A-site along the equivalent [100] directions [37,38]. However, the local [111] displacements are correlated in the medium range with the displacements along the [100] direction [71]. The measurement of X-ray absorption fine-structure (XAFS) revealed the off-centering of Nb ions in KTN at the B-site along the equivalent [111] direction at about 0.15 Å [21], while the off-centering of Li ions at the A-site along the [100] direction is expected to be 1.2 Å [38,104]. Since the off-center displacement of Li ions is much larger than that of Nb ions, the Li ion has a relatively larger dipole in comparison with the Nb ion. As a result, it is predicted that the correlations of displacements may lead along the equivalent [100] direction in Li-ion-rich KTN crystals. In the case of KLTN/0.025/0.38, the correlations of displacements are dominated by the Nb ions because of small concentrations of Li ions, and on average Li ions for the [100], [010], and [001] displacements are along an equivalent [111] direction [71,103], resulting in *A*_1_(*z*) mode of PNRs in 2.5%Li-doped KTN crystals. The invariance of the local mode of PNRs of the KLTN/2.5/38 crystal in comparison with that of the non-doped KTN indicates that the Li ions occupy only the A-site up to 2.5%Li-doping. Because of the off-center displacements of Li ions at the A-site, the doping of small amounts of Li ions in KTN may induce additional polarizations. These additional polarizations are cooperative with the neighboring PNRs, resulting in larger PNRs [27], and give rise to a weak RF due to the homovalent cations at both A- and B-sites [99]. On the other hand, the correlation of displacements between Li and Nb ions becomes stronger due to the relatively large Li concentrations in 5%Li-doped KTN; therefore, their displacements are predominantly along the [100], [010], and [001] directions. The switching of directions of polarization among [100], [010], and [001] may enhance the transverse polarization as shown in Figure 7c [69,103], resulting in the *E*(*x,y*) mode of PNRs in 5%Li-doped KTN.

ii.In 5%Li-doped KTN, another possibility for the change in the local *A*_1_(*z*) to the *E*(*x,y*) mode of PNRs may be the partial occupancy of a small percentage of the Li ions at the B-site. Wakimoto et al. [104] reported that the Li^+^–Li^+^ dipolar pairs in Ca-doped KLT/0.05 were reoriented with the reorientation of two nearest-neighbor Li^+^ ions. In 5%Li-doped KTN, it is predicted that the dipolar pairs may be present, and these dipolar pairs can be reoriented with the reorientation of the nearest neighbor Li^+^ ions. However, the interaction strength of dipole–dipole pairs depends on the magnitude of both dipoles, their orientations, and their proximity. In general, the proximity effect intensifies with an increase in the concentrations of dopants. Hence, the noticeable repulsive interaction between neighboring Li^+^ and dipolar pairs may exist in 5%Li-doped KTN owing to the proximity effect. The random orientations at high temperatures above *T** may cause the occupancy of a small percentage of the Li ions at the B-site through hopping from the A-site, as shown in Figure 8c. Thus, there is a noticeable interaction between Li and Nb ions in fluctuating PNRs corresponding to two sets of atomic displacements, giving rise to the *E*(*x,y*) mode of PNRs in 5%Li-doped KTN crystals. The *E*(*x,y*) mode of PNRs was also found in a relaxor PMN single crystal, in which off-center displacements of cations occurred at both A- and B-sites and the charge disorder existed only at the B-site [72].

In order to clarify the partial occupancy of Li ions at the B-site, we calculated the total energy, which is the key parameter for an equilibrium structure, of a unit cell using the density functional theory (DFT)-based CASTEP code [105]. Ye et al. [106] reported the occupancy of Sn ions at A- and B-sites from the energy point of view using first-principles calculations. XAFS measurements also revealed the occupancy of Mn ions at both A- and B-sites in Mn-doped SrTiO_3_ single crystals [107]. From the structural point of view, in non-doped KTN with a perovskite structure, the B-site is occupied by smaller Ta^5+^ (ionic radius 0.64 Å) and Nb^5+^ (ionic radius 0.64 Å) ions and the A-site is occupied by the larger K^1+^ ions (ionic radius 1.38 Å) [108], as shown in Figure 8a. Therefore, the RFs in non-doped KTN are too weak to show a relaxor-like nature [109], because B-site cations are homovalent and comparable in size and KTN possesses no charge disorder. Owing to the similar chemistry between K^1+^ and Li^1+^, the Li^1+^ (ionic radius 0.76 Å) would be expected to substitute for K^1+^ at the A-site in the Li-doped KTN crystal. However, the occupation of Li ions is slightly unfavorable due to the size consideration of the perovskite structure [110], and LiNbO_3_ is not a perovskite structure. In Ca-doped KTN, G. A. Samara and L. A. Boatner reported that the substitution of Ca occurs at the B-site by considering the ionic size [110]. Thus, there is a probability of partial occupancy of Li ions at the B-site in Li-doped KTN. For a better understanding of the partial occupancy of Li ions, the total energy per unit cell of the Li-doped KTN/0.27 was evaluated with first-principles calculations, as shown in Figure 8d–f. At first, the calculation of the total energy of the unit cell of KTN/0.27 was carried out by the Li-doping at the A-site, as shown in Figure 8d. As can be seen in Figure 8d, the total energy debases with Li-doping and shows a minimum when the Li content is 4.9% at the A-site. This result is a clear indication of the occupancy of Li ions at the A-site only in 2.5%Li-doped KTN crystal. Thus, the invariance of the local mode of PNRs in non-doped and 2.5%Li-doped KTN may be due to homovalent cations at both A- and B-sites. It is worth noting that the total energy starts to increase above the 4.9%Li-doping at the A-site (Figure 8e), which stimulates to study of the partial occupancy of Li ions at the B-site in 5%Li-doped KTN crystals. We calculated the energy of the Li occupancy at both A- and B-sites to clarify the partial occupancy of Li ions at the B-site. It is important to note that the total energy shows a more minimum value when the substitution level was 0.8% at the B-site and 4.2% at the A-site, as shown in Figure 8f. It is concluded that 0.8% of Li ions occupy the B-site. In 5%Li-doped KTN single crystals, the change in the local *A*_1_(*z*) to *E*(*x,y*) mode of PNRs is caused by the partial occupancy of Li ions at the B-site. The partial occupancy of Li ions at the B-site may enhance the RFs due to the charge disorder at the B-site as well. The enhanced RFs in addition to the growth of PNRs may induce the broadening of the elastic anomaly of 5%Li-doped KTN crystal in comparison with that of the non-doped KTN crystal.

### 3.4. Effects of Defects on PNRs in Li-Doped KTN

A schematic illustration of the composition gradient Li-doped KTN crystal wafer consists of dislocations that develop at the time of crystal growth along the [100] direction from a seed and the spatial distribution of *T*_C_ denoting the composition gradient are schematically shown in Figure 9a,b, respectively [58,111]. The dislocation density was high near the center (light yellow color) of the sample, that is, from the bottom to the top in Figure 9a. Therefore, the wafer was divided into two parts to observe the effects of defects on the composition gradient of Li-doped KTN. The “high defects”, represented with closed diamonds (blue), denote the region where the dislocation density is high, whereas “low defects”, marked with closed circles (red), indicate the region where the number density of dislocations is low. The defects’ origins might be essentially screw dislocations, and additionally protons or oxygen vacancies [112].

The positional dependence of elastic constants *C*_11_ (associated with LA phonon) and *C*_44_ (associated with TA phonon) of the Li-doped KTN wafer measured at room temperature (RT) is shown in Figure 9c,d. The variation in elastic constants with position reflects the composition gradient. These results are similar to those found in KTN single crystals with the ultrasonic pulse echo technique [35]. In the ferroelectric tetragonal phase at RT, the change in elastic constants in PZN–0.045PT and PZN–0.09PT crystals was studied by Ko et al. [43], and they hypothesized that the distinct behavior of the elastic constants was owing to the difference in the contribution from the domain walls and domain motion [43]. However, the change in elastic constants in Li-doped KTN was observed in the paraelectric cubic phase (Figure 9c,d), where there was no extrinsic contribution from the domain configurations. This fact suggests that the distinction of *C*_11_ and *C*_44_ with the position of the Li-doped KTN wafer can be caused by their different composition and the number density of defects. Another possibility is that there might be frequency dispersion within the Li-doped KTN wafer owing to the distinct number densities of PNRs and their correlation length depending mainly on the Nb content. In the high-defect regions, the acoustic phonons were considerably scattered by defects, so the sound attenuation increased, while velocity slightly decreased with the increase in defect densities, as shown in Figure 9c,d. The decrease in phonons frequency with the increase in the number density of defects suggests that defects may enhance the creation of PNRs. A similar change in elastic constants *C*_11_ and *C*_44_ with the number density of defects was reported in the case of non-ferroelectric materials such as silicon [45,46].

### 3.5. Effects of Electric Fields on PNRs in Li-Doped KTN

To understand the effects of the electric field on precursor dynamics, we investigated the 5%Li-doped KTN with *x* = 0.26 under zero-field cooling (ZFC) and an externally applied dc field using Brillouin scattering. The Brillouin spectra of the KLTN/0.05/0.26 single crystal at some selected temperatures recorded in a narrow frequency range under the ZFC process are shown in Figure 10a [113]. As can be seen in Figure 10a, the spectra are qualitatively similar to those observed in non-doped KTN in the paraelectric cubic phase. However, one interesting result observed in the ferroelectric phase is that the splitting of the TA mode occurs just below *T*_C_.

The frequency shift of the TA mode as a function of temperature is displayed in Figure 10b. The splitting of the TA mode indicates that the ferroelectric tetragonal phase of the Li-doped KTN crystals consists of two domain states. The high-frequency TA (HTA) mode stands for ferroelectric macro-domain states, whereas the low-frequency TA (LTA) denotes the random nanodomain states within macro-domain states, which were attributed to the freezing up of the polarization fluctuations of PNRs where the growth into macro-domains was blocked by the RFs [113]. Recently, Dey et al. [73] visualized randomly oriented nanodomains within macro-domain states in the ferroelectric phase of (1–*x*)Ba(Zr_0.2_Ti_0.8_)O_3_–*x*(Ba_0.7_Ca_0.3_)TiO_3_ (BZT–0.60BCT, *x* = 0.60) ceramics using piezo response force microscopy (PRFM). In the ferroelectric phase, the splitting of the acoustic modes due to the existence of multi-domain states was also found in typical RFEs [114].

The effect of the electric field in the ferroelectric phase of the KLTN/0.05/0.26 crystal was studied to discuss the microscopic origin of the splitting of the TA mode. The Brillouin scattering spectra of the KLTN/0.05/0.26 crystal measured under a constant temperature of 291 K at some selected electric fields applied along the [100] direction are shown in Figure 10c. The TA frequency shift as a function of electric fields under a constant temperature of 291 K is displayed in Figure 10d. It is important to note that the LTA and HTA modes become closer to each other with an increasing electric field along the [100] direction. At the critical electric field of *E*_C_ = 0.9 kV/cm, the split TA mode merges into a single peak, indicating the switching of the ferroelectric macro- and nanodomains along the direction of the applied field at *E*_C_. This result is a clear indication of the change of crystal from multi-domain to single domain state close to HTA by the suppression of pinning by RFs. Such a switching of multi-domain states into a single domain state under a moderate electric field was also observed in typical Refs [114].

A comparative study of the acoustic properties was performed between the field cooling (FC) and ZFC at 1.2 kV/cm to clarify the effect of electric fields on PNRs. The Brillouin scattering spectra of the KLTN/0.05/0.26 crystal observed under FC and ZFC at some selected temperatures are displayed in Figure 10e. As can be seen in Figure 10e, the TA mode splitting disappears under the FC at 1.2 kV/cm. Since the critical field of *E*_C_ = 0.9 is much smaller than the applied electric field of 1.2 kV/cm, the KLTN/0.05/0.26 crystal at 1.2 kV/cm mainly comprises a single domain state due to the complete switching of the random nanodomain states to the macro-domain state or single domain state resulting in the disappearance of splitting of the acoustic modes. Notably, the FWHM of the LA mode increases under FC in comparison with that of the ZFC process (Figure 10f). This means that the TA phonon under an electric field of 1.2 kV/cm was markedly scattered by PNRs, and the sound velocity may also decrease with the scattering. This result suggests that the electric field may enhance the creation and growth of PNRs. A significant decrease in the sound velocities with the growth and the number densities of PNRs was also observed in Li-doped KTN single crystal and wafer, respectively [27,111]. Under a moderate electric field of 1.2 kV/cm, the temperature interval between *T** and *T*_C_ was markedly increased (Figure 10f). The enhancement of the temperature interval between *T** and *T*_C_ can be attributed to the enhanced stability of PNRs [27], induced by the electric field along the [100] direction.

### 3.6. Critical End Point (CEP) and Injected Electrons in Li-Doped KTN

The study of temperature-field phase diagrams, including the critical end point (CEP), is one of the current topics in advanced materials such as ferroelectric materials owing to the giant electromechanical response near the CEP [115]. The CEP is the driving mechanism for the rotation of polarization and the associated giant electromechanical response of RFEs studied by M. Aftabuzzaman and S. Kojima [114]. Without injected electrons in 5%Li-doped KTN with *x* = 0.27, the temperature-field phase diagram concluding the CEP (*E*, *T*) = (3.3 kV/cm, *T*_C_ + 6 K) was investigated by Imai et al. [49]. In addition, a significant increase in the capacitance and permittivity of a Li-doped KTN crystal was observed using injected electrons [49]. However, injected electrons are trapped by the localized states in the crystal and stabilized. Therefore, a space-charge-induced electric field gradient is expected which significantly affects the CEP of the Li-doped KTN crystals. A study of the temperature dependence of the CP behavior was carried out to show the effect of injected electrons on the CEP. In this study, titanium electrode pairs were used to inject electrons into the KLTN/0.05/0.26 crystals by the application of voltages along the [100] direction [116]. The electric voltage dependence of the CP intensity at some selected temperatures of the KLTN/0.05/0.26 crystal in the paraelectric cubic phase is shown in Figure 11a [116]. Significant variation in the intensity of the CP was found at the critical voltages, reflecting the paraelectric cubic to the ferroelectric tetragonal phase transition. It is believed that most of the dynamic PNRs become static near *T*_C_ in RFEs. Therefore, the remarkable increase in the intensity of the CP at the critical voltage may be due to the alignment of randomly oriented dynamic and static PNRs along the direction of the applied voltage resulting in a [100] time-averaged polarization, and the crystal transforming into a lower tetragonal *P4mm* symmetry. A small hysteresis behavior of the CP intensity was observed in the paraelectric to ferroelectric phase transition at *V*_C1_ on heating and the ferroelectric to paraelectric phase transition at *V*_C2_ on cooling (Figure 11b). This fact indicates the slightly first-order nature of the ferroelectric phase transition. As can be seen in Figure 11b, the paraelectric to ferroelectric phase transition voltages, i.e., the critical electric voltages, were seen to shift to higher values with increasing temperature. However, the shift of *V*_C1_ is smaller than the shift of *V*_C2_; hence, *V*_C1_ and *V*_C2_ merge at around (*V,T*) = (190 V, *T*_C_ + 3.4 K), equivalent to (*E,T*) = (1.6 kV/cm, *T*_C_ + 3.4 K). This point in which the hysteresis behavior gradually diminished to zero was identified as the critical endpoint (CEP) of the KLTN/0.05/0.26 crystal. This point also denotes the change of the first-order to second-order nature of the ferroelectric phase transition [49]. Without electrons injected into the KLTN/0.05/0.27 crystal, the value of the CEP was found at (*E,T*) = (3.3 kV/cm, *T*_C_ + 6 K) [49]. The value of the critical electric field observed at the CEP of the KLTN/0.05/0.26 crystal with injected electrons is lower in comparison to that of the value found in KLTN/0.05/0.27 crystal with no injected electrons. Hence, the lowering of the average electric field of *E*_av_ = 1.7 kV/cm of the KLTN/0.05/0.26 crystal is due to the space-charge-induced electric field, which is generated by the injected and/or trapped electrons.

The temperature dependence of the dielectric properties of the KLTN/0.05/0.26 crystal was examined to investigate the effect of injected and/or trapped electrons on the CEP as well. The temperature-dependent real part of the dielectric constant (*Ɛ*’) under selected applied voltages is shown in Figure 11c. The dielectric anomaly was clearly seen at the paraelectric to ferroelectric phase transition temperature, *T*_C_ (Figure 11c). It is important to note that the slightly broad peak exhibited at about *T*_C_ at *V* = 0 V changes into a very sharp peak when approaching the CEP at *V* = 154.3V or *E* = 1.3 kV/cm. A sharp phase transition in the vicinity of the CEP was also observed in typical RFEs such as PMN-*x*PT [114,115]. It is also significant that the value of *T*_C_ is shifted to higher values in the range of 296.6–299.5 K under the application of applied voltage up to 178.0 V. It is possible to measure a very small area of a specimen to clarify the local heterogeneity using a combination of Brillouin scattering spectroscopy and optical microscopy [57]. However, the average value of the dielectric constant of the investigated specimen was obtained using conventional dielectric spectroscopy. Hence, the observed discrepancy of the CEP between dielectric and Brillouin spectroscopy measurements may be due to the field gradient induced by the injected and/or trapped electrons.

We also studied the position dependence of the Brillouin scattering spectra of the KLTN/0.05/0.26 crystal to clarify the effect of trapped electrons on the critical voltages. Figure 11g shows a schematic illustration of three positions in the crystal [116]. The electric voltage dependence of the CP intensity at three positions is shown in Figure 11d–f. The electron density is not uniform inside the crystal, and the electron density tends to be large near the cathode compared with the part near the anode when the electron injecting voltage is low. Hence, the absolute value of the injected-electron-induced electric field is almost zero at the cathode and it increases with the distance from the cathode and becomes the maximum at the anode. However, the crystal symmetry becomes tetragonal by an external electric field along the [100] direction because of the electrostriction. Hence, under a moderate electric field, the crystal part near the cathode tends to remain paraelectric (Figure 11g), whereas the part near the anode tends to become ferroelectric (Figure 11e) because the electric field is strong. It is worth noting that the value of the critical field near the center is somewhat smaller than that of the value near the anode (Figure 11d,e).

We also investigated the position-dependent critical electric field plot of the KLTN/0.05/0.26 crystal to understand the charge distribution inside the crystal. It is challenging to rationalize the actual behavior of the electric field inside the crystal because of the limited number of observation positions. For simplicity, we consider the positions *x* = *d* (close to anode), *x* = *d*/2 (close to center), and position *x* = 0 (close to cathode), where *x* is the distance from the cathode and *d* is the thickness of the sample. The detailed calculation procedure of the position-dependent critical electric field was reported in Ref. [116]. Figure 11h shows the position dependence of the critical field of the KLTN/0.05/0.26 crystal. The position-dependent electric field exhibits a slight deviation from linear behavior based on the results of the position dependence of the intensity of the CP (Figure 11h). This result suggests that the deviation of the electric field from linear behavior could be due to an inhomogeneous distribution of charges injected into the crystal with applied voltages. A difference between the calculated and observed capacitance owing to the inhomogeneous distribution of charges was also reported in Ref. [49]. It is also important to note that the CP intensity near the center was slightly smaller than that of the CP intensity near the anode (Figure 11d,e). As the electric field is strong enough near the anode in comparison with the other parts of the crystal, the randomly oriented dynamic and static PNRs may start to align along the direction of the applied field near the anode, and bring about an intense CP near the anode.

## 4. Conclusions

The understanding of precursor dynamics associated with polar clusters called PNRs is crucial for the enhancement of the functionality of ferroelectric materials. The precursor dynamics in relaxor-like ferroelectric non-doped KTN and Li-doped KTN crystals near the paraelectric to ferroelectric phase transition temperate, *T*_C_, were investigated using Brillouin scattering. A significant softening of sound velocity and a remarkable increase in sound attenuation in the vicinity of *T*_C_ were observed in both non-doped KTN and Li-doped KTN single crystals. A significant increase in the intensity of the central peak (CP) and sound attenuation as well as a remarkable softening of the transverse sound velocity, *V*_TA_, were clearly found below the intermediate temperature, *T**, denoting the beginning of the rapid growth of PNRs. In both crystals, the temperature evolution of the average size of dynamic PNRs is evaluated, and below *T** it markedly increases towards the *T*_C_. The relaxation time, which was estimated from the width of a broad central peak (CP) of non-doped KTN and Li-doped KTN crystals, exhibits a critical slowing down towards *T*_C_, indicating the order–disorder behavior of a ferroelectric phase transition.

The effects of Li-doping on PNRs in KTN crystals were clearly observed by the broadening of the elastic anomaly and the stretching of the slowing down with weak random fields (RFs) related to PNRs in the Li-doped KTN crystals in comparison with the non-doped KTN one. A significant increase in the average size of the PNRs towards *T*_C_ was also found with Li-doping, which indicates a broadening and extension of the temperature region of the elastic anomaly. The expansion of the temperature region and broadening of the elastic anomaly could indicate the enhanced relaxor nature of Li-doped KTN single crystals. The analysis of dielectric properties confirms the enhanced relaxor nature of Li-doped KTN crystals. A change of local *A*_1_(*z*) to *E*(*x,y*) modes of PNRs was observed in 5%Li-doped KTN crystal. A first-principles calculation based on density functional theory (DFT) revealed that 4.2% of Li ions occupy the A-site and 0.8% of Li ions occupy the B-site in 5%Li-doped KTN. The heterovalence charge disorder at the B-site induced by the partial occupancy of Li ions could be the microscopic origin of the change in the local mode of PNRs.

The co-existance of macro-domain and random nanodomains associated with PNRs in the ferroelectric phase of Li-doped KTN was observed by the splitting of the TA mode. The switching of nano- and macro-domain states into a single-domain state was found at a critical field of *E*_C_ = 0.9 kV/cm. An increase in number density and growth of PNRs under a moderate electric field was suggested by a significant increase in the damping of the acoustic mode in comparison with that of the ZFC process. The defect induces different number densities of PNRs found in Li-doped KTN wafer as well.

The temperature-field phase diagram including the critical end point (CEP) of Li-doped KTN crystal was clarified. The trapped electrons’ effects were evidenced by the lower value of the CEP in comparison with the Li-doped KTN crystal with no trapped electrons in the literature. The difference in CEP between the dielectric result averaged over all areas of a sample and the Brillouin result at a fixed small area revealed the inhomogeneous distribution of injected and/or trapped electrons in Li-doped KTN single crystal. The inhomogeneous distribution of injected and/or trapped electrons was confirmed by the observation of position dependence of the CP intensity.

## Figures and Tables

**Figure 1 materials-16-00652-f001:**
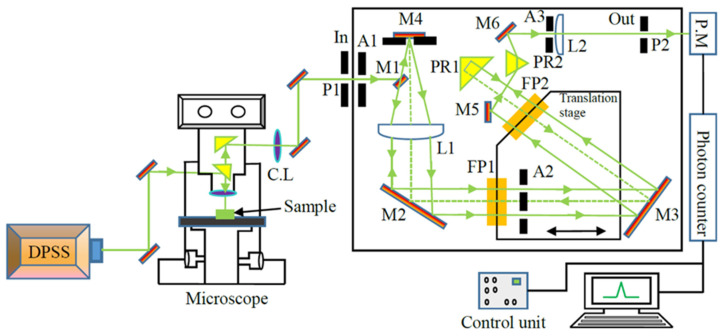
A schematic illustration of a Sandercock 3 + 3 pass tandem interferometer equipped with a photomultiplier and optical microscope [57,58].

**Figure 2 materials-16-00652-f002:**
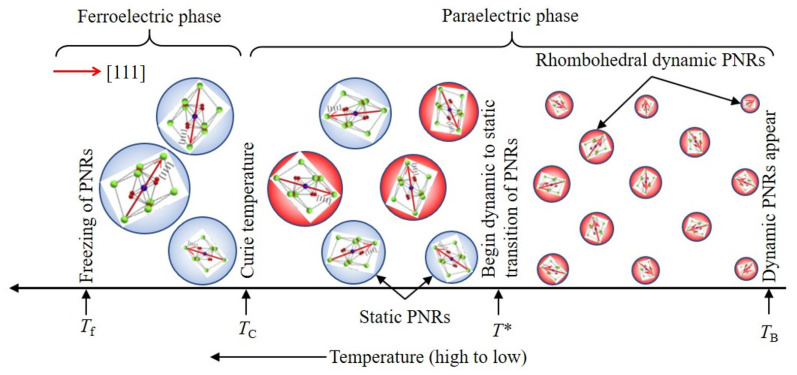
A schematic illustration of the temperature evolution of PNRs in relaxor-like KTN crystals based on the inelastic light scattering results. Red denotes the dynamic PNRs and blue indicates static PNRs, where the arrow denotes the orientation of PNRs. The dynamic PNRs appear at the so-called Burns temperature, *T*_B_. Upon cooling, the size of the dynamic PNRs increases slowly and dynamic PNRs become more sluggish due to the cooperative interaction with the neighboring PNRs. Eventually, the dynamic PNRs start to become static PNRs at the intermediate temperature, *T**. For further cooling, the size of PNRs becomes intense due to their growth in addition to the merger of neighboring PNRs.

**Figure 3 materials-16-00652-f003:**
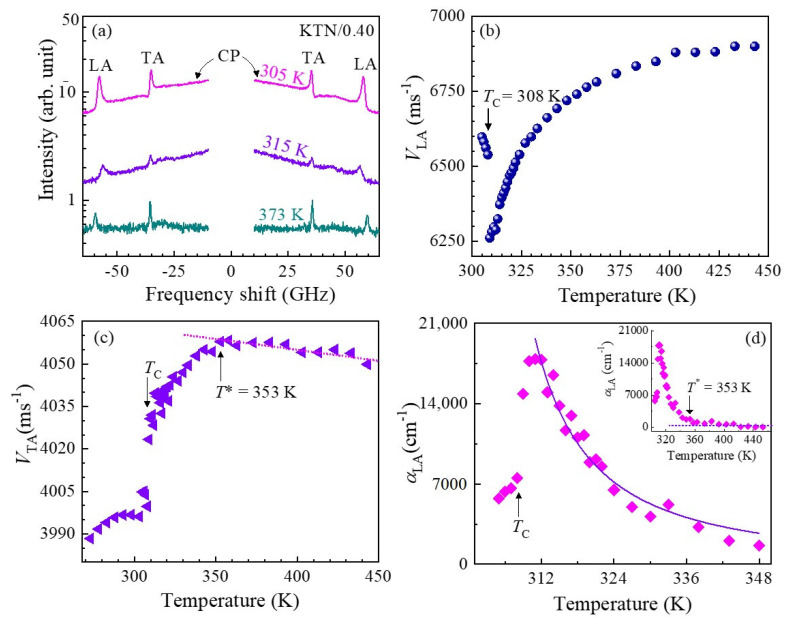
(**a**) The observed Brillouin spectra of non-doped KTN/0.40 single crystal at some selected temperatures measured with FSR = 75 GHz. (**b**) The longitudinal velocity of the LA mode, and (**c**) the transverse velocity of the TA mode in a KTN/0.40 single crystal as a function of temperature. (**d**) The temperature dependence of the longitudinal sound attenuation in a KTN/0.40 single crystal. The short dashed and solid lines in (**c**) and (**d**) are the guide to eyes and the best fit result obtained using Equation (7), respectively. The inset in figure (**d**) exhibits the plot on a large scale to denote the *T**. The data were adapted from Refs. [25,58].

**Figure 4 materials-16-00652-f004:**
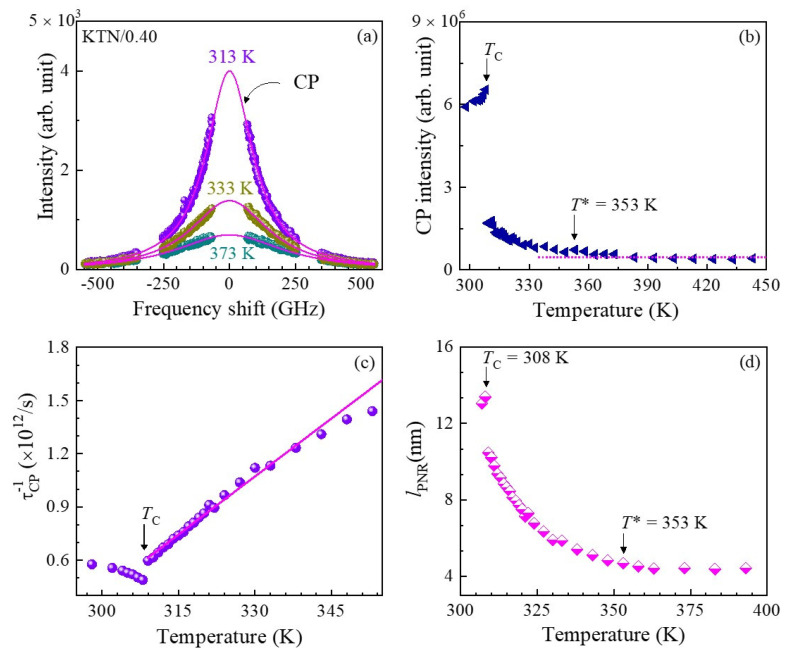
(**a**) Brillouin scattering spectra the KTN/0.40 crystal at selected temperatures measured with FSR = 300 GHz, where the solid lines denote the curves fitted using the Voigt function. (**b**) The CP intensity of the KTN/0.40 crystal as a function of temperature. (**c**) The temperature dependence of the inverse of the relaxation time determined from the CP width, where the solid line is the result of best fit using Equation (9). (**d**) The mean size of the dynamic PNRs of KTN/0.40 single crystal as a function of temperature. The data were adapted from Refs. [25,58].

**Figure 5 materials-16-00652-f005:**
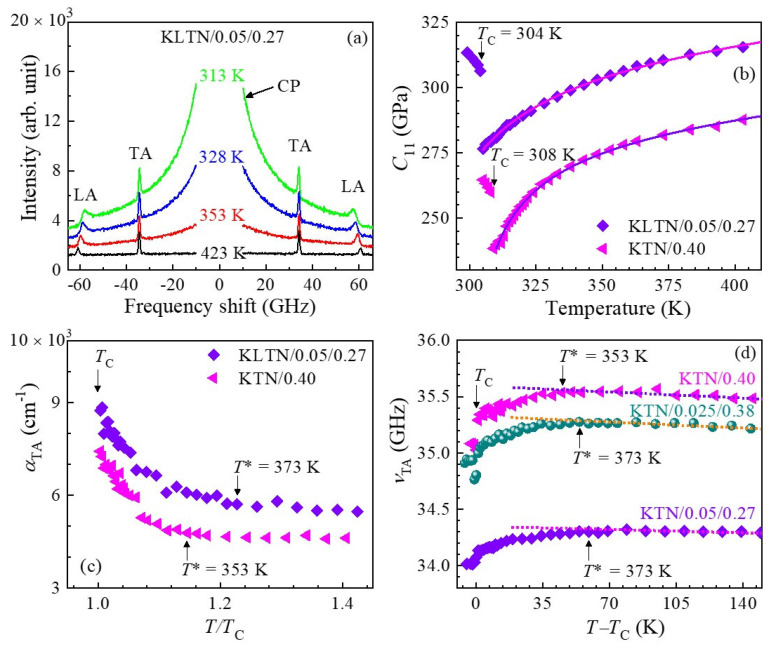
(**a**) Brillouin scattering spectra of the KLTN/0.05/0.27 single crystal at some selected temperatures recorded with FSR = 75 GHz. (**b**) Temperature-dependent elastic constant *C*_11_ of both KTN/0.40 and KLTN/0.05/0.27 single crystals. The solid lines are the best fit result obtained from Equation (11) (**c**) The attenuation of the TA mode in the KTN/0.40 and KLTN/0.05/0.27 crystals as a function of *T/T_C_*. (**d**) The observed frequency shift of the TA mode of KTN/0.40, KLTN/0.025/0.38, and KLTN/0.05/0.27 crystals as a function of normalized temperature. The short dashed lines are guides to the eyes. The data were adapted from Refs. [27,58].

**Figure 6 materials-16-00652-f006:**
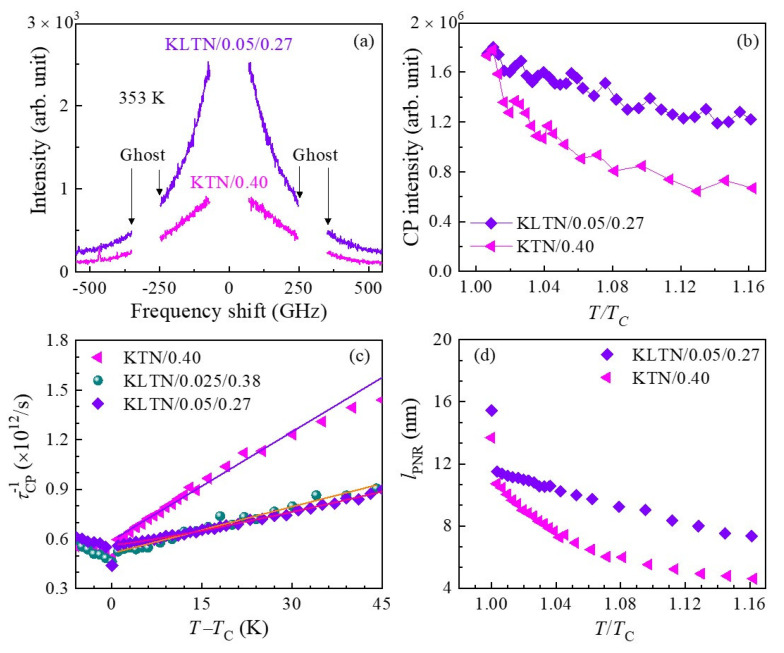
(**a**) The CP spectra of KTN/0.40 and the KLTN/0.05/0.27 single crystals measured with the FSR = 300 GHz at 353 K. (**b**) The CP intensity of KTN/0.40 (solid diamonds) and KLTN/0.05/0.27 (solid triangles) single crystals as a function of *T/T*_C_. (**c**) The inverse of the relaxation time estimated from the width of the CP of non-doped KTN, 2.5%Li-doped KTN, and 5%Li-doped KTN single crystals as a function of normalized temperature. The solid lines are the lines fit with Equation (12) (**d**) The size of a dynamic PNR of non-doped KTN and 5%Li-doped KTN crystals as a function of *T*/*T*_C_. The data were adapted from Refs. [27,58].

**Figure 7 materials-16-00652-f007:**
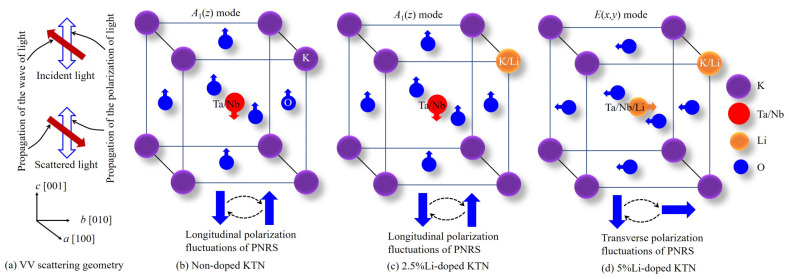
(**a**) A typical depiction of the VV scattering geometry. A schematic interpretation of the polarization fluctuations of PNRs caused by *A*_1_(z) mode in (**b**) non-doped KTN, (**c**) 2.5%Li-doped KTN, and *E*(*x,y*) mode in (**d**) 5%Li-doped KTN crystals observed in the VV scattering under the condition of fixed A-site cations [103].

**Figure 8 materials-16-00652-f008:**
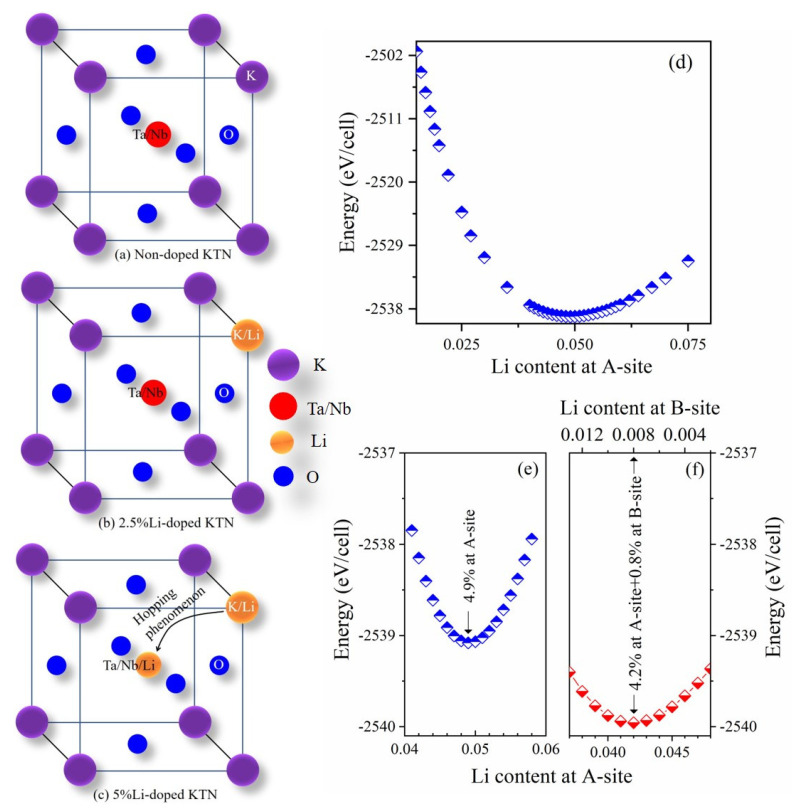
A schematic illustration of the lattice showing the location of substituents in the cubic perovskite, (**a**) non-doped KTN, (**b**) 2.5%Li-doped KTN, and (**c**) 5%Li-doped KTN single crystals, where a small percentage of Li ions occupy the B-site in 5%Li-doped KTN and the Li ions only occupy the A-site in 2.5%Li-doped KTN. The total energy per formula unit obtained using the density functional theory (DFT) calculation as a function of the Li content (**d**) at the A-site (**e**) near the 5%Li-doping at the A-site and (**f**) at A- and B-sites of the Li-doped KTN/0.27 single crystal. The data were adapted from Ref. [103].

**Figure 9 materials-16-00652-f009:**
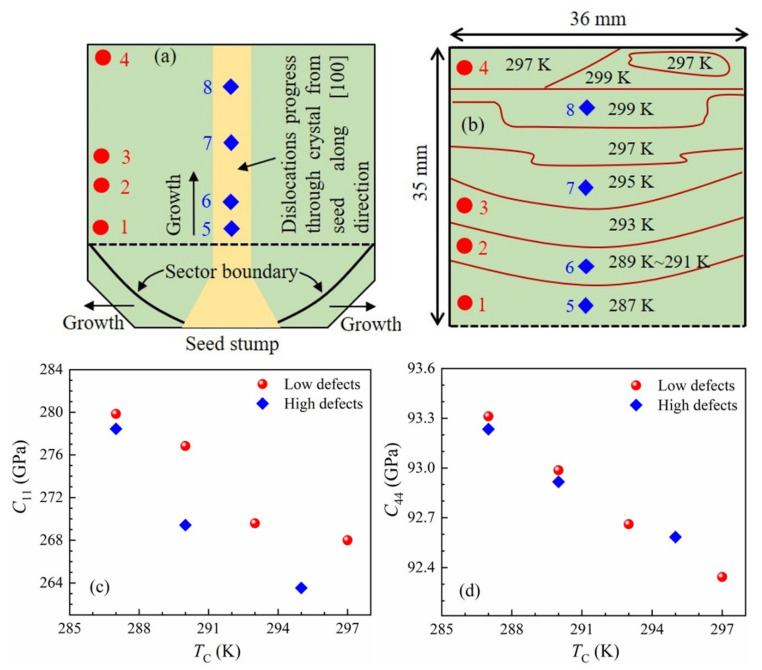
Schematic illustration of (**a**) spatial heterogeneity and (**b**) spatial dependence of *T*_C_ in a composition gradient on Li-doped KTN wafer. Diamonds (blue) and closed circles (red) denote the regions with high and low defect density, respectively. The elastic constants (**c**) *C*_11_ and (**d**) *C*_44_ of a composition gradient Li-doped KTN wafer as a function of position. The data were adapted from Refs. [58,111].

**Figure 10 materials-16-00652-f010:**
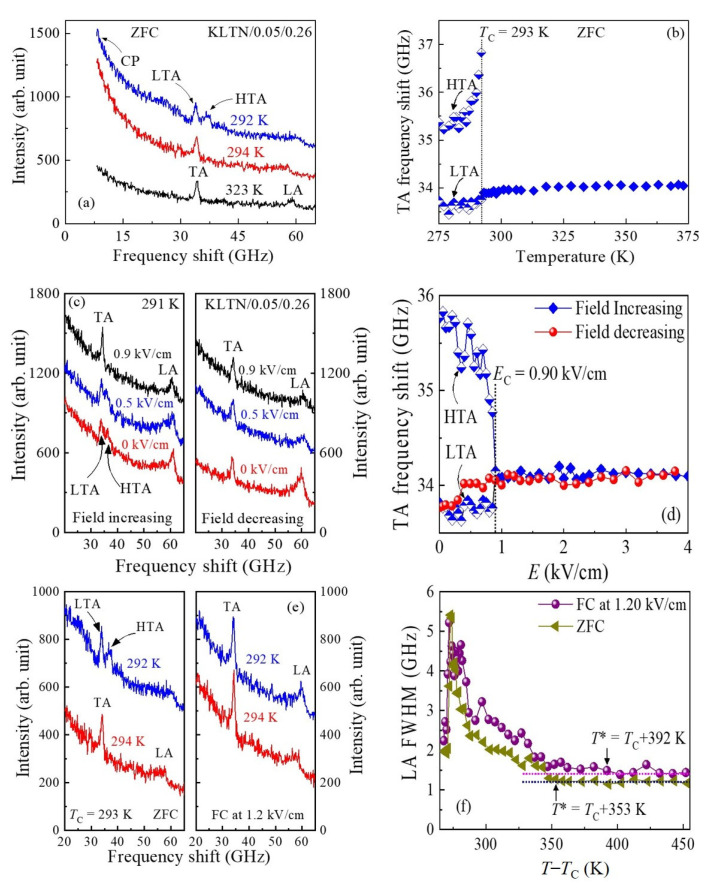
(**a**) Brillouin scattering spectra of the KLTN/0.05/0.26 single crystal at some selected temperatures measured at zero field cooling (ZFC). (**b**) The temperature dependence of the frequency shift of the TA mode of the KLTN/0.05/0.26 single crystal. (**c**) The Brillouin scattering spectra of the KLTN/0.05/0.26 crystal at some selected electric fields measured in the ferroelectric phase under a constant temperature of 291 K, where the electric field was applied along the [100] direction. (**d**) The frequency shift of the TA mode observed in the ferroelectric phase as a function of electric fields. (**e**) The typical Brillouin spectra of the KLTN/0.05/0.26 crystal at some selected temperatures measured under ZFC and field cooling (FC) at 1.20 kV/cm. (**f**) The temperature dependence of the FWHM of the LA mode observed under ZFC and FC process [113].

**Figure 11 materials-16-00652-f011:**
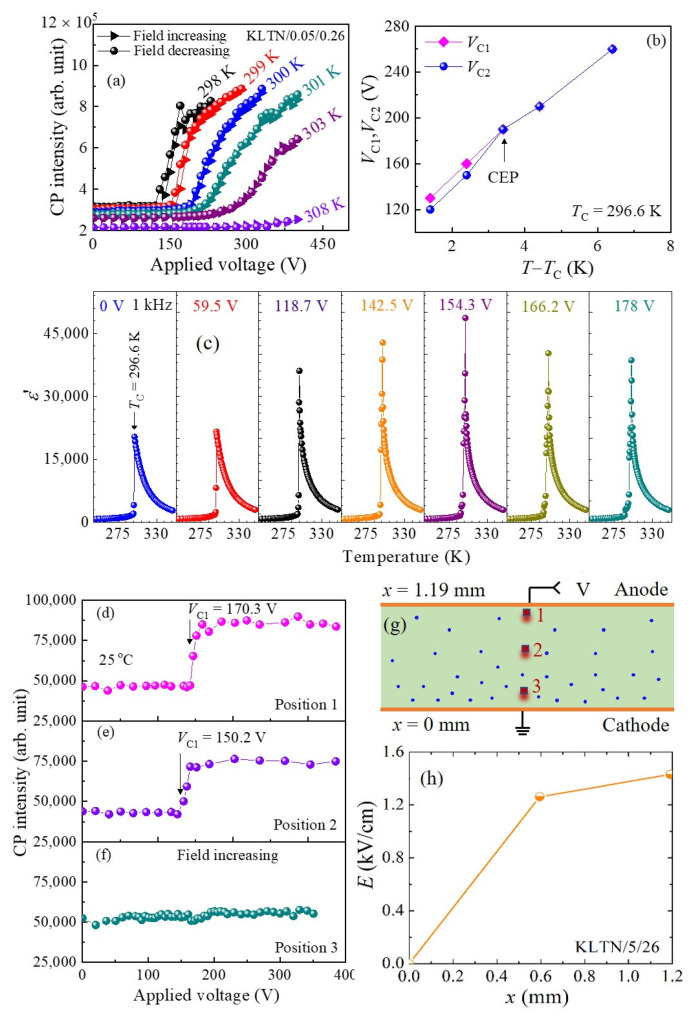
(**a**) The CP intensity of the KLTN/0.05/0.26 crystal at some selected temperatures as a function of applied voltage. (**b**) Temperature dependence of critical electric voltages of a phase transition, where *V*_C2_ denotes the transition from ferroelectric to paraelectric and *V*_C1_ indicates the transition from paraelectric to ferroelectric phase. (**c**) The dielectric constants of the KLTN/0.05/0.26 crystals measured at 1 kHz under the DC biasing voltages as a function of temperature. The electric voltage dependence of the observed CP intensity at (**d**) close to the anode (position 1), (**e**) close to the center (position 2), and (**f**) close to the cathode (position 3). The three positions in the KLTN/0.05/0.26 crystal are schematically drawn in (**g**). (**h**) The critical electric field of the KLTN/0.05/0.26 crystal as a function of position. The data were adapted from Ref. [116].

## Data Availability

Not applicable.

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
