# Peer review of "Brillouin Scattering Study of Electro-Optic KTa1−xNbxO3 Crystals"

_materials, 2023, doi:10.3390/ma16020652_

Round 1

Reviewer 1 Report

This Review reports Brillouin scattering study of KTN crystals. The paper is well written except for some minor issues.

1. Figure 3d partially covers the description of the axis in Figure 3b.

2. In Figure 4, unify the size of the letters in the axis description (4a and 4d are smaller).

3. The place of Tc in Figure 2 is confusing to me. What is the definition of Tc according to the authors? From what measurements can it be determined for materials such as relaxor ferroelectrics and relaxor-like ferroelectrics ?

4. The characteristics of the tested crystals are missing - the method of growing, etc.

5. Line 357: What is the definition of the order parameter according to the authors? We think about materials like relaxor ferroelectrics and relaxor-like ferroelectrics.

6. Complete the explanations of the symbols in formula (7).

7. How to understand the electric field-induced TC in Abstract section? Is it about shifting the phase transition temperature under the influence of the electric field?

8. How dislocations density was determined in Figure  9?

9. In Figure 11c it is not visible whether there is a shift of the permittivity peak. It is desirable to indicate it. We can only see differences in the max value. Are the measurements on the same sample all the time? How long the electric field acts on the crystal can be important.

10. What technique was used to measure E at different sample points (Figure 11h)?

11. Do Figures 11 e)-g) indicate charge segregation under the influence of the E-field? What charges and in which direction do they move?

Reviewer 2 Report

The object of research of the present manuscript is KTa1-xNbxO3 crystals by inelastic light scattering spectroscopy. The manuscript is well structured and the applied methods used are appropriate. The authors have tracked the temperature dependence of the transverse acoustic (TA) and the longitudinal acoustic (LA) modes of KTN/0.40 single crystal. The influence of polar nanoregions (PNRs) on the obtained dependences.

At the next stage is the last influence of Li-doping effects on PNRs in KTN and electro-optic effects in KLTN crystals.

Therefore, I think that the article is appropriate and can be published in its current form.

Reviewer 3 Report

In this paper, the author's reviews mainly focus on recent progress in understanding precursor dynamics in Nb-rich KTN crystals studied by the Brillouin scattering. The intense central peak and significant softening of sound velocity are observed above the Curie temperature due to the polarization fluctuations in PNRs. The effects of Li-doping, defects, and electric field on the growth and/or creation of PNRs are found by changes in acoustic properties. The electric field-induced TC including critical endpoint and field gradient by trapped electrons are discussed as well. Which is one of the current interests in material science. This article is clear, concise, and suitable for the scope of the journal. Several small suggestions are supplied:
1. Suggest the authors supply one table to collect recent achievements on this topic.
2. The authors use many graphs, suggesting the authors check the copy issue.
3. Suggest the authors enhance the introduction part with more latest references.
